# Sleep Patterns, Excessive Daytime Sleepiness, and Sleep Disturbance Among First Nations Children in Saskatchewan

**DOI:** 10.3390/clockssleep7020021

**Published:** 2025-04-25

**Authors:** Chandima P. Karunanayake, Charlene Thompson, Meera J. Kachroo, Donna C. Rennie, Warren Seesequasis, Jeremy Seeseequasis, James A. Dosman, Mark Fenton, Vivian R. Ramsden, Malcolm King, Sylvia Abonyi, Shelley Kirychuk, Niels Koehncke, Punam Pahwa

**Affiliations:** 1Canadian Centre for Rural and Agricultural Health, University of Saskatchewan, 104 Clinic Place, Saskatoon, SK S7N 2Z4, Canada; meera.kachroo@usask.ca (M.J.K.); donna.rennie@usask.ca (D.C.R.); jccquasis22@gmail.com (J.S.); james.dosman@usask.ca (J.A.D.); shelley.kirychuk@usask.ca (S.K.); niels.koehncke@usask.ca (N.K.); pup165@mail.usask.ca (P.P.); 2College of Nursing, University of Saskatchewan, 104 Clinic Place, Saskatoon, SK S7N 2Z4, Canada; charlene.thompson@usask.ca; 3Community A, P.O. Box 96, Duck Lake, SK S0K 1J0, Canada; wseesequasis@beardysband.com; 4Department of Medicine, University of Saskatchewan, Royal University Hospital, 103 Hospital Drive, Saskatoon, SK S7N 0W8, Canada; mef132@mail.usask.ca; 5Department of Academic Family Medicine, University of Saskatchewan, West Winds Primary Health Centre, 3311 Fairlight Drive, Saskatoon, SK S7M 3Y5, Canada; viv.ramsden@usask.ca; 6Department of Community Health & Epidemiology, College of Medicine, University of Saskatchewan, 107 Wiggins Road, Saskatoon, SK S7N 5E5, Canada; malcolm.king@usask.ca (M.K.); sya277@mail.usask.ca (S.A.)

**Keywords:** First Nations, children, sleep, screen time, daytime sleepiness

## Abstract

Sleep is essential for maintaining both mental and physical well-being. It plays a critical role in the health and development of children. This study investigates sleep patterns and habits of First Nations children, the prevalence of sleep disturbances, and excessive daytime sleepiness (EDS), along with the factors associated with EDS. Our 2024 First Nations Children Sleep Health Study assessed the sleep health of children aged 6 to 17 years living in a First Nation in Canada. Statistical analyses were performed using comparison tests and logistic regression models. A total of 78 children participated; 57.7% were boys. The average age of the participants was 10.49 years (SD = 3.53 years). On school days, children aged 6 to 9 years slept an average of one additional hour, while on weekends, they slept an extra 40 min compared to adolescents aged 10 to 17 years. Only 39.7% of the children (ages 6 to 17) slept alone in a room, with more than 80% of the children sharing a bed every night. Only 30.6% of the children aged 6 to 9 years and 7.2% of the adolescents aged 10 to 17 years adhered to the recommended maximum screen time of 2 h on school days. More than two-thirds of the children reported experiencing sleep disturbances. The prevalence of EDS was 19.7%. After adjusting for age and sex, it was determined that the children who snored loudly and those who did not sleep in their own beds were more likely to experience abnormally high levels of daytime sleepiness. A high proportion of children exceeded the recommended screen time, an important public health issue. Further, identifying sleep patterns among children will facilitate the diagnosis and treatment of disordered sleep.

## 1. Introduction

Sleep is a condition of decreased awareness and activity that is crucial for sustaining both mental and physical well-being [1]. It plays a crucial role in the health and well-being of children [1,2,3] and is vital for their growth, cognitive development, emotional well-being, and behavior [4]. The recommended range for optimal sleep varies with age, according to the standards of the American Academy of Sleep Medicine. For children aged 6–12 years, the optimal sleep recommendation is 9–12 h, while for teenagers aged 13–18 years, it is 8–10 h [5]. Furthermore, sleep disruption has been linked to various chronic health conditions, including obesity, diabetes, and cardiovascular disease [6,7,8,9], as well as influencing childhood cognition and behavior [10]. Research indicates that sleep habits are linked to the academic performance and overall well-being of both non-Indigenous and Indigenous children [3,4,11,12,13,14]. Furthermore, poor sleep patterns contribute to negative health outcomes for children [15,16].

The availability and nighttime use of electronic entertainment and communication devices have been linked to short sleep duration, obesity, poor diet quality, and lower levels of physical activity among Canadian children [9]. Another study in Canada found that total sedentary time was negatively associated with sleep duration [17]. Additionally, a population-based study reported that longer sleep durations were associated with a lower risk of obesity, better diet quality, and increased physical activity levels [18]. Meeting sleep recommendations was associated with higher odds of positive psychosocial health among Canadian children aged 5 to 13 years and youth aged 14 to 17 years [19]. Moreover, adhering to screen time guidelines was also linked to greater odds of positive psychosocial health among youth [19]. Approximately 68% of Canadian school-aged children aged 10 to 13 years and 72% of adolescents aged 14 to 17 years sleep for the recommended amount per night on average across the week [20]. This means that nearly one-third of Canadian children and adolescents are not getting the recommended amount of sleep.

Excessive daytime sleepiness (EDS) is a common issue among children (10% to 20%) and adolescents (16% to 47%) [21]. A lack of sufficient sleep at night is associated with increased daytime sleepiness, which can negatively impact attention, motivation to learn, and academic performance [1,22,23,24,25]. Various studies have identified several factors associated with EDS, including socioeconomic status [26], sex [27], smoking [28], caffeine consumption [29], physical activity [30], screen time [31], and depression [32].

Numerous studies have investigated sleep patterns in Indigenous Australian children. These studies have found a significant association between late sleeping habits, shorter sleep duration, and body mass index [33,34,35]. Additionally, one study reported that 32% of Indigenous children experienced difficulties with initiating and maintaining sleep, while 20% faced challenges with sleep–wake transitions, and another 20% reported excessive daytime sleepiness [36]. A systematic review on sleep health among Indigenous Australian children revealed that 20% experienced severe daytime sleepiness, 10.9% had short sleep durations, 50% were identified as late sleepers, and 14.2% reported snoring [2].

However, there have been limited studies exploring the sleep habits and sleep duration of Indigenous children in Canada. So far, we have identified only one article that discusses sleep duration among Indigenous youth [37] and another that examines the sleep habits of Indigenous children [38] in Canada. The qualitative study conducted by Hovey et al. [38] reported that prolonged screen time, unhealthy meals, a generational gap, and a lack of outdoor physical activities negatively affected children’s sleep. Parents and teachers also mentioned issues such as inattentiveness, misbehavior, irritability, fatigue, and children falling asleep at their desks [38]. This indicates a significant information gap in this area of research. Our objectives were to explore and examine the sleep patterns and habits of First Nations children in Canada, the prevalence of sleep disturbances and EDS, and the factors associated with EDS.

## 2. Results

### 2.1. Descriptives

In the study, 78 children participated, of which 45 (57.7%) were boys, and 33 (42.3%) were girls. The ages of the participants ranged from 6 to 17 years, with one adolescent who had recently turned 18 also included in the study and grouped into the 10–17 years age group. The average age of the participants was 10.49 years (SD = 3.53 years). Age distribution was as follows: 25 children (32.1%) were aged 6–8 years, 29 children (37.2%) were aged 9–12 years, and 24 children (30.8%) were aged 13 years and older. Approximately half of the children fell within the normal BMI category (49.4%), while 27 children (35.1%) were classified as obese.

### 2.2. Sleep Duration

Figure 1 illustrates the differences in children’s sleep patterns and duration on school days compared to weekend days. On school nights, 27% of the children reported going to bed after 10 p.m., whereas 62.8% indicated that they went to bed after 10 p.m. on weekends. A similar trend was observed in the wake-up times: 88.5% of the children reported waking up between 6 a.m. and 8 a.m. on school days, but only 15.4% did so on weekends. Additionally, we found that 70.5% of the children slept for more than nine hours over the weekend, compared to 65.4% who achieved the same amount of sleep on school nights.

This study revealed notable differences in the circadian rhythms of children and adolescents on weekdays compared to weekends. On both school days and weekends, a greater number of adolescents tended to go to bed after 10 p.m. While their waking times remained consistent during school days, many adolescents woke up later on weekends. Furthermore, the proportion of adolescents who slept more than 9 h on both weekdays and weekends was lower than that of children aged 6 to 9 years. On average, the children slept an additional hour on school days and an extra 40 min on weekends compared to the adolescents. (Table 1).

### 2.3. Sleep Habits, Sleep Patterns, and Screen Time

Less than one-third of the participants (28.2%) reported that they did not wake up in the middle of the night. Many children struggled to get out of bed in the morning, with 23.1% stating that they always had difficulty, and 47.4% saying they sometimes did. Additionally, 51.3% of the children took less than 20 min to fall asleep. Furthermore, three-quarters of the participants indicated that their children, or they themselves, got enough sleep at night. Only 39.7% of the children woke up independently, while 26.9% needed someone to wake them up. Additionally, only 39.7% of the children slept alone in a room, while more than 80% of the children slept in the same bed every night (Table 2).

There were notable differences in sleep habits and patterns between children and adolescents. The adolescents tended to wake up in the middle of the night more frequently than the children. Additionally, a higher percentage of adolescents took over an hour to fall asleep compared to their younger counterparts. Many adolescents reported not getting enough sleep and often waking up on their own. More than half of the adolescents slept alone and did not share their bed with anyone else. Moreover, a greater number of adolescents tended to sleep in the same bed every night compared to the children (Table 2).

During a school day, 17.9% of the participants reported spending 2 hours or fewer on screens using a phone, other mobile devices, or a computer, while 24.3% spent 8 or more hours on screens (see Figure 2a). Additionally, in a typical week over the past 3 months, 19.2% reported watching television or videos for 2 hours or fewer each day, whereas 21.8% spent 8 or more hours (see Figure 2b). On weekends, 6.5% reported spending 2 hours or fewer on screen on their devices, while 40.3% spent 8 or more hours (see Figure 2a). In the same timeframe, 6.5% typically watched television or videos for 2 hours or fewer per day, whereas 46.8% spent 8 or more hours (see Figure 2b).

About two-fifths of the children spent 8 h or more on screens each day during the weekend. Almost half of them typically spent 8 h or more watching TV or videos during weekends over the past three months. However, this trend was lower on school days, with only 24.3% of the children spending time on screens, and 21.8% watching TV or videos (Figure 2a,b).

Table 3 illustrates the daily hours spent on screens, as well as the time spent watching TV or videos, during school and weekend days for the study children (ages 6–9) and adolescents (ages 10–17). On weekends, both children and adolescents averaged 8 h or more of screen time and television viewing, which is significantly higher than on school days. However, the adolescents exhibited a much higher overall screen time compared to the children on both school days and weekends. While the amount of time spent watching television and videos was similar for the two groups on school days, the adolescents watched more television and videos on weekends.

Additionally, the report indicated that 48.7% of the children “always” and 32.1% “sometimes” engaged in activities such as playing video games, watching TV, surfing the internet, or sending texts within one hour before going to bed. During an average school week, 20.5% of the children “always” watched TV in bed, while 33.3% “sometimes” did so. Furthermore, 32.1% of them “always” used a smartphone in bed, and 17.9% “sometimes” used one. Only a small percentage of children (2.6%) woke up “always” in the middle of the night due to receiving texts from friends, while 11.5% of the children woke up “sometimes” for the same reason.

### 2.4. CASC Results

Out of the 78 children, 67 completed the CASC questionnaire, which includes 24 questions about sleep disturbances. These questions cover a range of issues, including bedtime problems, sleep breathing, unstable sleep, parasomnia, sleep movements, and daytime problems. The total score for the 24 questions ranges from 0 to 72. The mean CASC score was 22.55, with a standard deviation of 9.55. There were no significant mean differences in the CASC scores between the sexes (*p* = 0.121). Among the 67 respondents, 68.7% (46 out of 67) reported a CASC sleep disturbance score of 18 or higher, indicating that they experienced sleep problems. This means that more than two-thirds of the children in this population reported having sleep issues.

### 2.5. ESS-CHAD Results

Out of the 78 children, 66 completed the ESS-CHAD questionnaire with 8 questions. The mean ESS-CHAD score was 7.38, with a standard deviation of 4.22. There were no significant differences in the mean ESS-CHAD scores (7.08 (±4.31 SD) for boys versus 7.81 (±4.12 SD) for girls) between the sexes (*p* = 0.486). Among the 66 respondents who completed the ESS-CHAD questionnaire, 19.7% (13 out of 66) reported a score greater than 10, indicating excessive daytime sleepiness problems. There was no significant difference in the mean ESS-CHAD scores between children (7.52 ± 4.26) and adolescents (7.26 ± 4.24), with a *p* value of 0.806. The prevalences were also not significantly different (19.4% vs. 20% for children and adolescents, respectively; *p* = 0.948).

A higher proportion of boys, in the 9 to 12 years age group, in the obese BMI category, and in the parental highest educational level of “Some university/bachelor’s degree/professional certificate/graduate degree” category reported experiencing excessive daytime sleepiness. Children who had sleep disturbances, snored loudly, did not sleep in their own beds, frequently used a smartphone while in bed, and those who lived in homes with visible mold were more likely to report excessive daytime sleepiness (Table 4). However, in a multiple logistic regression analysis, excessive daytime sleepiness (as measured by the Epworth Sleepiness Scale) was not significantly related to factors such as age, sex, obesity, screen time (specifically, watching TV or videos during school days), parental highest educational level, or housing conditions. The study found that excessive daytime sleepiness was significantly associated with loud snoring and the practice of not sleeping in one’s own bed. Specifically, after adjusting for age and sex, the results showed that children who snored loudly were 9.5 times more likely to report excessive daytime sleepiness, while those who did not sleep in their own beds were six times more likely to experience abnormally high levels of daytime sleepiness. The study did not identify significant correlations between excessive daytime sleepiness and other factors such as sleep disturbances, total sleep hours, physical activity, or screen time with technological devices (Table 4).

## 3. Discussion

In this study, approximately two-fifths of the children reported spending 8 h or more on screens daily on weekends. Nearly half of them typically spent 8 h or more watching TV or videos on weekends over the past three months. Conversely, this trend was lower on school days, with only 24.3% of the children spending time on screens, and 21.8% watching TV or videos. Additionally, the children’s sleeping patterns varied between school and weekend days; adolescents exhibited significantly higher overall screen time compared to younger children across both periods. The amount of time spent watching television and videos was also higher on weekends. During weekends, the children tended to sleep and wake up later compared to school days. However, three-quarters (75.6%) of the participants indicated that their children, or they, get enough sleep at night. On average, children aged 6–9 years were found to sleep one hour more than adolescents (10–17 years).

In this study, approximately 74% of the adolescents slept 8 or more hours on school days compared to 76% on weekends. In contrast, 89% of the children aged 6–9 years slept more than 9 h on school days, while 78% did so on weekends. It is important to note that the recommended sleep durations vary across different age groups for adolescents [5,40,41], which is why direct comparisons were not made. Furthermore, the Statistics Canada 2014–2015 Canadian Health Measures Survey (CHMS) indicated that about 74% of children and youth aged 5–17 years in Canada meet the sleep duration recommendations outlined in the Canadian 24-Hour Movement Guidelines for Children and Youth [42]. The Canadian Health Behaviour in School-Aged Children Study (HBSC) found that children and adolescents typically sleep about 1 h more on weekends than on weekdays [20]. In line with this, our study observed that the surveyed children slept an average of 1 h and 15 min more on weekends, while the adolescents slept 1 h and 40 min more compared to school days. It is normal for an adolescent’s circadian clock to shift to a later schedule, which conflicts with the school system’s requirement for them to be up and ready by 8:30 a.m. or even 9 a.m. As a result of this natural shift, the adolescents in this study often reported feeling tired and struggling to wake up, which is a typical aspect of their development.

A previous study on the sleep habits of Indigenous children [38] reported that the negative influence of technology has hindered children from achieving adequate sleep. Some contributing factors include the late-night use of cell phones, computers, televisions, and video games, as children have easy access to the internet and their devices. Hovey et al. also noted that technology in the bedroom is challenging to monitor and control [38]. The overuse of technology not only leads to sleep deprivation but also results in a loss of imagination and creativity. The Canadian guidelines recommend limiting the recreational screen time to no more than 2 h per day for children aged 5 to 17 years [40]. However, in Canada, only 29% of the children meet these recommendations [42]. According to data from three cycles (2009–2015) of the Canadian Health Measures Survey (CHMS), Bang et al. reported that only 32.6% of children aged 5 to 11 years and 24.7% of youth aged 12 to 17 years met the recommendation of maximum 2 h of screen time [19]. In the current study, it was found that only 30.6% of First Nations children aged 6 to 9 years and 7.2% of First Nations adolescents aged 10 to 17 years adhered to the maximum 2 h screen time guideline on school days. These proportions were observed to be lower during weekends.

There are limited published studies examining sleep habits and patterns in children. However, Owens et al. [43] reported findings on the sleep habits of elementary school children from kindergarten through fourth grade. They noted that 26.2% of the children rarely fell asleep within 20 min, while 41.7% of the children aged 6 to 9 years in our study reported that they took more than 20 min to fall asleep. When asked about their sleep duration, 11.7% of the children in the Owens study reported usually feeling that they slept too little. In contrast, only 2.8% of the children aged 6 to 9 years in our study indicated that they felt that they were not getting enough sleep. Regarding night wakings, 14.6% of the children in the Owens study responded that they usually woke up at night while their parents believed they were asleep [43]. In our study, however, 58.4% of the children in the same age group reported waking up in the middle of the night at least once. Owens et al. also found that 28.1% of their study children usually had difficulty waking up or getting out of bed, whereas our study indicated that 63.8% of the participating 6-to-9-year-olds experienced trouble getting out of bed in the morning [43]. In our study, the percentages of adolescents aged 10 to 17 years who experienced trouble getting out of bed in the morning were significantly higher. Previous studies have suggested that cultural differences exist regarding sleep habits and the amount of sleep that school-aged children typically achieve [43,44,45,46]. However, because our sample was small and there were slight differences in the wording of the questions, we cannot conclusively attribute these variations in sleep habits solely to First Nations children.

Using the CASC sleep disturbance score, where a score of 18 or higher indicates sleep issues, more than two-thirds (68.7%) of the children in this study reported experiencing sleep problems. Excessive daytime sleepiness (EDS) was assessed using the ESS-CHAD questionnaire, and about one-fifth (19.7%) of the children reported EDS. This prevalence is very similar to that in the adult population (19.8%) in a study [47]. Additionally, there were no differences in the mean ESS-CHAD scores between the sexes. Previous studies have reported the prevalence of EDS in children and adolescents to range from 10% to 47%, depending on the definitions used [21,43,46,48,49,50,51,52]. Consistent with our findings, Australian Indigenous children also reported a high rate of severe daytime sleepiness, at 20% [2]. In our study, 18.2% of the children reported loud snoring, a percentage slightly lower than that of 14.2% reported for Australian Indigenous children living in the community [2]. Notably, the children who snored loudly and did not sleep in their own beds were more likely to experience EDS. Specifically, those who snored loudly were 9.5 times more likely to report EDS, and those who did not sleep in their own beds were six times more likely to report EDS. This could be attributed to differences in sleep environment, safety, and reduced sleep disturbances when sleeping in a familiar bed. Although this study observed a high screen time, no links were established between screen time and daytime sleepiness. In contrast, other studies have demonstrated an association between screen time and daytime sleepiness [53,54].

### Strengths and Limitations

This study is one of the first to examine sleep patterns, sleep disturbances, and EDS among First Nations children aged 6 to 17 years using quantitative measures. However, one limitation is the small sample size, which affects the representativeness of the children’s population. It is important to recognize this limitation when interpreting results and generalizing study findings to other First Nations populations. Retrospective power analyses [55] indicated that this study was moderately powered to detect a difference between the population proportion (p0) and the sample proportion (p1) of 0.103 (where p0 = 0.300, and p1 = 0.197) in the prevalence of EDS, using a type I error rate (α) of 0.05. A larger sample size may have enhanced the robustness of the results. Children’s sleep is influenced by various factors, including socioeconomic status, cultural activities, and family traditions, which were not included in this study. Due to the study’s cross-sectional design, we cannot draw long-term conclusions based solely on the presented results. Nonetheless, while causality cannot be definitively established, this study provides valuable insights into the sleep habits and sleep patterns of Canadian First Nations children.

## 4. Materials and Methods

### 4.1. Data Collection and Study Protocol

The First Nations Sleep Health Study examined the sleep health of individuals aged six years and older who live in either a Willow Cree or a Woodland Cree First Nation community in rural Saskatchewan, Canada [56]. This analysis focused on data collected from a baseline cross-sectional study conducted in 2024, specifically among children in one Cree First Nation community. All children aged 6–17 years, members of participating reserves and living on reserve and in nearby towns, who were members of the reserves and attending on-reserve schools, were eligible to participate, whereas children not attending on-reserve schools were not be eligible to participate in this phase of the research. Ethical approval for the children’s study was granted by the University of Saskatchewan’s Biomedical Research Ethics Board (REB) (Bio Certificate #18-110). The detailed research protocol used for the children’s study is provided below.

School officials approved the study and its associated surveys. The survey was in paper form and distributed to parents through community schools, along with parental consent and child assent forms for clinical testing. The clinical assessments conducted at the schools included anthropometric measurements. The parents returned the questionnaires and the consent and assent forms to the schools, where they were collected by the research team. A CAD 35.00 incentive was offered to those who returned their questionnaires.

After receiving a Certificate of Approval from the University of Saskatchewan’s Biomedical REB and in order to help parents, caregivers, and the community understand potential concerns related to the children’s survey and assessments, the research team held meetings with the directors of education and individuals responsible for the education portfolios in the community. An information session for parents and caregivers was organized at each school site before beginning the survey data collection. An Elder from the community was invited to attend this session and offer valuable advice for conducting the project in a culturally sensitive way. During the information session, details about the survey, as well as the consent and assent processes, were discussed. Information regarding the upcoming survey, along with details about parent, caregiver, and child involvement, was also shared through the school newsletter.

The school leadership granted permission to distribute cross-sectional paper survey packages to the parents of students in grades 1–12. These packages included questionnaires for the parents of children in grades 1–5, as well as consent and assent forms for clinical testing for students in grades 1–12. For grades 6–12, the surveys were conducted during classroom time in a group setting, arranged at the convenience of the schools. Students in grades 1 to 12 underwent clinical assessments of height, weight, and waist circumference in a private area of the school. A research assistant conducted all clinical assessments within the schools and oversaw the distribution of the study packages to the students and/or parents.

The approach outlined above has been successfully utilized by our research team, which includes members of the communities, in previous studies with the communities [57,58]. This method is efficient for recruitment, while minimizing disruptions to the school environment. However, during this study, only 17 students were able to participate on school grounds. To address the issue of low participation and increase the response rate, we organized a community event during the summer break to gather data from children aged 6 to 13 years and from those aged 14 to 17 years. The families attending the community event received a study package upon arrival. This package included a cover letter, parent/child information, consent/assent forms, and a questionnaire. The questionnaire was designed to be completed either by a parent or by a caregiver for children aged 6 to 13 years, and by a parent/caregiver or by the child themselves for those aged 14 to 17 years. The participants were encouraged to fill out the surveys at designated tables and chairs provided at the event venue. For those who consented to have a clinical assessment, these assessments were conducted in a designated private area. Screens were set up to ensure privacy for the participants and allow researchers or research assistants to take measurements discreetly.

### 4.2. Anthropometric Measurements

Each participant’s standing height, sitting height, weight, and waist circumference (abdominal girth) were measured. All measurements were taken by trained researchers or research assistants to ensure accuracy and reliability. Height was measured against a wall in centimeters (cm) using a fixed tape measure, while weight was measured in kilograms (kg) with a spring scale. The children were weighed and measured while wearing indoor clothing and in stocking feet. Body mass index (BMI) was calculated using the standard formula weight (kg) divided by height (cm) squared. Based on age- and sex-standardized BMI, the children were classified as obese, overweight, or normal/underweight, following the guidelines set out by Cole et al. [59,60].

### 4.3. Sleep Duration

To determine the average sleep hours, we calculated the mean of sleep hours on school days and weekends, categorizing the results into two groups, i.e., less than 9 h and 9 h or more, considering the recommendation standards from the American Academy of Sleep Medicine [5] for the regression analysis.

### 4.4. Sleep Habits and Sleep Patterns and Screen Time

The survey included several variables related to sleep habits, as follows: how many times a child wakes up in the middle of the night; how long it takes for the child to fall asleep; satisfaction with the amount of sleep that the child gets at night; who wakes the child up in the morning; whether the child has trouble getting out of bed in the morning; how many people share a room with the child; whether the child sleeps in the same bed every night; and whether the child has their own bed. Additionally, the survey included questions about screen time, including the time spent using screens (such as phones, other mobile devices, or computers); the average time spent watching television or videos in a typical week over the past three months; how often the child plays video games, watches television, browses the internet, or sends texts within one hour before going to sleep; the average frequency of watching television in bed during a school week; the average frequency of using a smartphone in bed during a school week; and how often the child wakes up in the middle of the night due to receiving texts from friends.

### 4.5. Child and Adolescent Sleep Checklist (CASC)

The CASC is designed to identify sleep habits and screen for sleep problems among preschoolers, elementary school children, and high school students [61]. The CASC Sleep Disturbance Score is calculated based on responses to a checklist of 24 questions. Each response is scored using a 4-point frequency scale: a score of 3 indicates “Always”, 2 indicates “Usually”, 1 indicates “Occasionally”, and 0 indicates “Never/Do not know” (for the parent version only). The total score from the 24 questions can range from 0 to 72. Children with a CASC Sleep Disturbance Score of 18 or higher are considered to have sleep problems.

### 4.6. Epworth Sleepiness Scale (ESS) for Children and Adolescents (ESS-CHAD)

The ESS-CHAD is a modified version of the Epworth Sleepiness Scale (ESS) that has been validated for measuring daytime sleepiness in children and adolescents [62,63,64,65,66]. Further, item 8 “sitting and eating a meal” was replaced with “doing homework or taking a test” [67]. The ESS-CHAD assesses the degree of sleepiness, with a score that ranges from 0 to 24. Using a 4-point ordinal response scale, children and adolescents rate how likely they are to doze off in eight different situations. A high score indicates great daytime sleepiness. An ESS-CHAD score greater than 10 is considered abnormal and suggests excessive daytime sleepiness (EDS) [63,65].

### 4.7. Statistical Analysis

Statistical analyses were conducted using SPSS version 29 (IBM Corp. Released 2024; IBM SPSS Statistics for Windows, Version 29.0, Armonk, NY, USA: IBM Corp.). The data obtained from self-reported questionnaires completed by parents or children were used to assess sleep habits, daytime sleepiness, and sleep disturbances. Descriptive statistics, including mean, median, and standard deviation (SD), are reported for continuous variables. The *p*-value from an independent samples *t*-test was determined for comparisons of sample means. For categorical variables, frequencies and percentages are provided. Chi-squared tests were employed to assess the bivariate associations between outcomes and the independent variables of interest. When cell counts were small, Fisher’s exact test *p*-values are reported. Although some analyses were conducted by age group (children aged 6–9 years and adolescents aged 10–17 years [39]) and noted in the results, regression analysis was performed on the entire sample. Multiple logistic regression models were utilized to predict the relationship between the binary outcome of the Epworth Sleepiness Scale (ESS) classification (normal/abnormal) and a set of explanatory variables. A series of logistic regression models were fitted to identify whether potential risk factors, confounders, and interaction effects significantly contributed to predicting the ESS scores. Based on the bivariate analysis, variables with a *p*-value of less than 0.20 and less than 25% missing information were considered candidates for the multiple logistic regression model. All statistically significant variables (*p* < 0.05), as well as important clinical factors (age and sex), were included in the final multiple regression model. Interactions between potential effect modifiers were examined and retained in the final model if the *p*-value was less than 0.05.

## 5. Conclusions

The prevalence of EDS observed in this study is comparable to that found for other Indigenous groups. A high proportion of children exceeded the recommended screen time, an important public health issue. Parents play a crucial role in implementing screen time interventions and establishing bedtime routines for their children. These interventions encourage children to reduce their time spent in front of the TV and other stationary screens, such as computers, video games, and mobile devices like smartphones and tablets. Instead, children should engage in physical activities and have limited screen time. Additionally, parents can benefit from education on how to create healthy bedtime routines. It is also important to minimize noise in the house during bedtime to promote better sleep. Further, identifying sleep patterns among children will facilitate the diagnosis and treatment of disordered sleep.

## Figures and Tables

**Figure 1 clockssleep-07-00021-f001:**
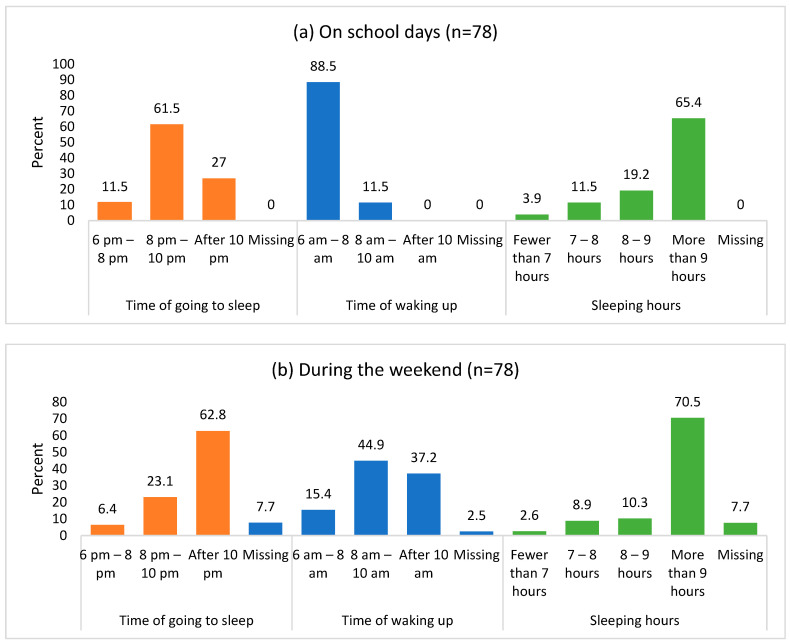
Distribution of sleep patterns and duration of sleep for children during (**a**) school days and (**b**) weekend days (n = 78 children).

**Figure 2 clockssleep-07-00021-f002:**
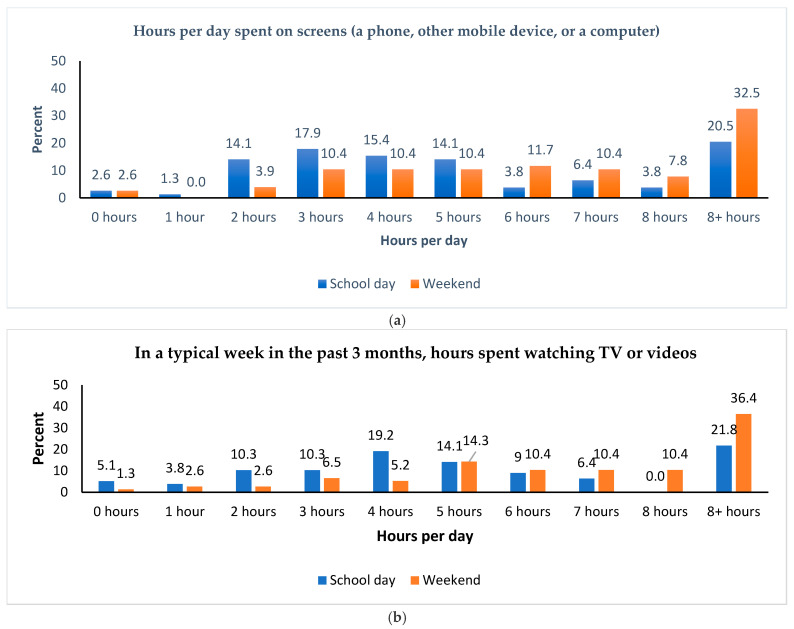
(**a**) Hours per day spent on screens during school days and weekend days. (**b**) In a typical week in the past 3 months, hours spent watching TV or videos during school days and weekend days.

**Table 1 clockssleep-07-00021-t001:** Difference in sleep duration for children and adolescents between school days and weekend days (n = 36 children and 42 adolescents *).

Time/Duration		On School Days n (%)		On the Weekend n (%)	
		Children (6–9 Years)	Adolescents (10–17 Years)	Children (6–9 Years)	Adolescents (10–17 Years)
Time of going to sleep	6 p.m.–8 p.m.	3 (8.3)	6 (14.3)	1 (2.8)	4 (9.5)
	8 p.m.–10 p.m.	29 (80.6)	19 (45.2)	15 (41.6)	3 (7.1)
	After 10 p.m.	4 (11.1)	17 (40.5)	18 (50.0)	31 (73.8)
	Missing	-	-	2 (5.6)	4 (9.5)
Time of waking up	6 a.m.–8 a.m.	31 (86.1)	38 (90.5)	7 (19.4)	5 (11.9)
	8 a.m.–10 a.m.	5 (13.9)	4 (9.5)	17 (47.2)	18 (42.9)
	After 10 a.m.	-	-	11 (30.5)	18 (42.9)
	Missing	-	-	1 (2.8)	1 (2.4)
Sleeping hours	Fewer than 7 h	-	3 (7.1)	-	2 (4.8)
	7–8 h	1 (2.8)	8 (19.1)	3 (8.3)	4 (9.5)
	8–9 h	3 (8.3)	12 (28.6)	3 (8.3)	5 (11.9)
	More than 9 h	32 (88.9)	19 (45.2)	28 (77.8)	27 (64.3)
	Missing	-	-	2 (5.6)	4 (9.5)
	Mean ± SD^#^ in hours	10.21 ± 0.87	9.14 ± 1.91	11.47 ± 3.17	10.81 ± 3.26

* According to the definition of the World Health Organization (WHO), adolescence is the phase of life between childhood and adulthood, from ages 10 to 19 [39]. In this table, 6–9-year-old participants were considered children, and 10–17-year-old participants were considered adolescents. SD^#^—standard deviation.

**Table 2 clockssleep-07-00021-t002:** Frequency of sleep habits and patterns among children and adolescents *.

Item	Total (n = 78) (6–17 Years) n (%)	Children (n = 36) (6–9 Years) n (%)	Adolescents (n = 42) (10–17 Years) n (%)
On average, how many times does your child/do you wake up in the middle of the night?			
Never	22 (28.2)	11 (30.6)	11 (26.2)
One time	29 (37.2)	14 (38.9)	15 (35.7)
2–3 times	18 (23.0)	5 (13.9)	13 (31.0)
More than 3 times	5 (6.5)	2 (5.6)	3 (7.2)
Missing	4 (5.1)	4 (11.1)	-
On average, how long does it take your child/you to fall asleep?			
20 min or less	40 (51.3)	21 (58.3)	19 (45.2)
40 min or less	17 (21.8)	7 (19.4)	10 (23.8)
1 h or less	9 (11.5)	4 (11.1)	5 (11.9)
More than 1 h	12 (15.4)	4 (11.1)	8 (19.0)
Amount of sleep at night			
Getting enough sleep	59 (75.6)	34 (94.4)	25 (59.5)
Not getting enough sleep	16 (20.5)	1 (2.8)	15 (35.7)
Missing	3 (3.8)	1 (2.8)	2 (4.8)
On average, waking up in the morning:			
Wakes up by him/herself	31 (39.7)	12 (33.3)	19 (45.2)
Wakes up by him/herself with an alarm clock	6 (7.7)	2 (5.6)	4 (9.5)
Has to be awaken by someone	21 (26.9)	7 (19.4)	14 (33.3)
Varies	20 (25.6)	15 (41.7)	5 (11.9)
How many times in an average week, having trouble getting out of the bed in the morning			
Never	17 (21.8)	10 (27.8)	7 (16.7)
Sometimes (1–3 times)	37 (47.4)	15 (41.6)	22 (52.4)
Always (4 times and over)	18 (23.1)	8 (22.2)	10 (23.9)
Missing	6 (7.7)	3 (8.3)	3 (7.1)
When going to sleep or going to bed, how many people are sharing the same room with your child/you			
Alone	31 (39.7)	7 (19.4)	24 (57.1)
1 person	19 (24.4)	11 (30.6)	8 (19.0)
2 persons	18 (23.1)	10 (27.8)	8 (19.0)
3 or more persons	9 (11.5)	8 (22.2)	1 (2.4)
Missing	1 (1.3)	-	1 (2.4)
Does your child/Do you have their/your own bed			
Yes	64 (82.1)	26 (72.2)	38 (90.5)
No	12 (15.4)	10 (27.8)	2 (4.8)
Missing	2 (2.6)	-	2 (4.8)
Does your child/Do you sleep in the same bed every night?			
Yes	65 (83.3)	28 (77.8)	37 (88.1)
No	11 (14.1)	8 (22.2)	3 (7.1)
Missing	2 (2.6)	-	2 (4.8)

* According to the definition of the World Health Organization (WHO), adolescence is the phase of life between childhood and adulthood, from ages 10 to 19 [39]. In this table, 6–9-year-old participants were considered children, and 10–17-year-old participants were considered adolescents.

**Table 3 clockssleep-07-00021-t003:** Percentage of hours per day spent on screens and watching TV or videos during school days and weekend days by children (6–9 years) and adolescents (10–17 years).

Hours per Day	Hours per Day Spent on Screens (a Phone, Other Mobile Devices, or a Computer)				In a Typical Week in the Past 3 Months, Time Spent Watching Television or Videos in a School Day or a Weekend Day			
	Children (6–9 Years)		Adolescents (10–17 Years)		Children (6–9 Years)		Adolescents (10–17 Years)	
	School Day	Weekend Day	School Day	Weekend Day	School Day	Weekend Day	School Day	Weekend Day
0 h	2.80	5.70	2.40	0.00	0.00	0.00	9.50	2.40
1 h	2.80	0.00	0.00	0.00	8.30	5.70	0.00	0.00
2 h	25.00	5.70	4.80	2.40	11.10	5.70	9.50	0.00
3 h	19.40	17.10	16.70	4.80	8.30	2.90	11.90	9.50
4 h	19.40	11.40	11.90	9.50	25.00	8.60	14.30	2.40
5 h	8.30	11.40	19.00	9.50	8.30	14.30	19.00	14.30
6 h	8.30	14.30	0.00	9.50	8.30	11.40	9.50	9.50
7 h	5.60	11.40	7.10	9.50	8.30	14.30	4.80	7.10
8 h	0.00	8.60	7.10	7.10	0.00	8.60	0.00	11.90
8+ h	8.30	14.30	31.00	47.60	22.20	28.60	21.40	42.90

**Table 4 clockssleep-07-00021-t004:** Crude association of ESS-CHAD score (ESS-CHAD > 10) with important factors. Results presented in terms of column percentages, unadjusted odds ratio (OR) estimates, and 95% confidence intervals (CI) (n = 66).

Variable/Factor	Total n (%)	Epworth Sleepiness Score		Fisher Exact Test *p* Value ^±^	Unadjusted OR and (95% CI)	Adjusted OR ^#^ and (95% CI)
		Normal (ESS-CHAD ≤ 10) n (%)	Abnormal (ESS-CHAD > 10) n (%)			
Sex						
Male	39 (59.1)	31 (58.5)	8 (61.5)	1.000	1.14 (0.33, 3.94)	0.83 (0.18, 3.83)
Female	27 (40.9)	22 (41.5)	5 (38.5)		1.00	1.00
Age group						
6–8 years	20 (30.3)	17 (32.1)	3 (23.1)	0.582	1.00	1.00
9–12 years	27 (40.9)	20 (37.7)	7 (53.8)		1.98 (0.44, 8.88)	1.76 (0.26, 12.06)
13 years and older	19 (28.8)	16 (30.2)	3 (23.1)		1.06 (0.19, 6.05)	1.25 (0.13, 12.49)
Body mass index						
Normal	34 (51.5)	29 (54.7)	5 (38.5)	0.199	1.00	-
Overweight	11 (16.7)	10 (18.9)	1 (7.7)		0.58 (0.06, 5.58)	
Obese	21 (31.8)	14 (26.4)	7 (53.8)		2.90 (0.78, 10.78)	
Parent’s highest education level						
Less than high school	30 (45.5)	24 (45.3)	6 (46.2)	0.232	1.00	-
High school completed/technical college/trade certificate	15 (22.7)	14 (26.4)	1 (7.7)		0.29 (0.03, 2.62)	
Some university/bachelor’s degree/professional certificate/graduate degree	13 (19.7)	8 (15.1)	5 (38.5)		2.5 (0.59, 10.46)	
Not stated	8 (12.1)	7 (13.2)	1 (7.7)		0.57 (0.06, 5.58)	
Sleep problems						
CASC Score ≥ 18	40 (67.8)	29 (61.7)	11 (91.7)	0.043	6.83 (0.81, 57.45) *	-
CASC Score < 18	19 (32.2)	18 (38.3)	1 (8.3)		1.00	
Snore loudly						
Yes	12 (18.2)	7 (13.2)	5 (38.5)	0.049	4.11 (1.04, 16.19) **	9.45 (1.61, 55.34) **
No	54 (81.8)	46 (86.8)	8 (61.5)		1.00	1.00
Sleep in own bed						
Yes	54 (83.1)	46 (86.8)	8 (66.7)	0.194	1.00	1.00
No	11 (16.9)	7 (13.2)	4 (33.3)		3.29 (0.78, 13.864)	7.36 (1.06, 51.05) **
Average sleep hours ^&^						
Less than 9 h	9 (14.8)	8 (16.0)	1 (9.1)	1.000	0.53 (0.06, 4.69)	-
9 h or more	52 (85.2)	42 (84.0)	10 (90.9)		1.00	
Smoke in the house						
Yes	19 (28.8)	15 (28.3)	4 (30.8)	1.000	1.13 (0.30, 4.22)	-
No	47 (71.2)	38 (71.7)	9 (69.2)		1.00	
Any water leakage or water damage in house						
Yes	19 (28.8)	15 (28.3)	4 (30.8)	1.000	1.13 (0.30, 4.22)	-
No	47 (71.2)	38 (71.7)	9 (69.2)		1.00	
Visible mold indoors						
Yes	8 (12.1)	5 (9.4)	3 (23.1)	0.185	2.88 (0.59, 14.06)	-
No	58 (87.9)	48 (90.6)	10 (76.9)		1.00	
Physical activities in last 7 days						
Yes	54 (85.7)	44 (88.0)	10 (76.9)	0.375	0.46 (0.10, 2.13)	-
No	9 (14.3)	6 (12.0)	3 (23.1)		1.00	
Screen time during a school day						
2 or less hours	14 (21.2)	12 (22.6)	2 (15.4)	0.718	1.00	-
More than 2 h	52 (78.8)	41 (77.4)	11 (84.6)		1.61 (0.31, 8.28)	
Watching TV or videos during a school day in a typical week in the past 3 months						
2 or less hours	15 (22.7)	12 (22.6)	3 (23.1)	1.000	1.00	-
More than 2 h	51 (77.3)	41 (77.4)	10 (76.9)		0.98 (0.23, 4.12)	
On average school week, child uses a smartphone in bed						
Always	20 (30.3)	14 (26.4)	6 (46.2)	0.234	3.21 (0.78, 12.23)	3.79 (0.54, 26.53)
Sometimes	12 (18.2)	9 (17.0)	3 (23.1)		2.50 (0.47, 13.31)	1.66 (0.24, 11.72)
Occasionally/Never/Do not know	34 (51.5)	30 (56.6)	4 (30.8)		1.00	1.00

* *p* < 0.10; ** *p* < 0.05. ^#^ Adjusted for age and sex in the multivariable model. ^&^ Average sleep hours calculated taking the average of school day sleep hours and weekend sleep hours. ^±^ Fisher’s exact test calculator: https://www.quantitativeskills.com/sisa/statistics/five2hlp.htm accessed on 20 January 2025.

## Data Availability

The First Nations community own and control the data and the data release as per the research agreements with the communities and the OCAP principles (https://fnigc.ca/ocap-training/, accessed on 16 July 2024). Requests for data access can be made to the Chief and Council of the community at reception@beardysband.com.

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
