# Peer review of "Sleep Patterns, Excessive Daytime Sleepiness, and Sleep Disturbance Among First Nations Children in Saskatchewan"

_2624-5175, 2025, doi:10.3390/clockssleep7020021_

Round 1
Reviewer 1 Report
Comments and Suggestions for Authors
This paper focuses on the sleep research of First Nations children in Canada. Although it has certain research value, there are deficiencies in aspects such as relevant experimental considerations, mechanism analysis, and relevant recommendations. My main concerns are as follows:
- The content in the Introduction section is too simplistic. Please conduct an in-depth analysis of the background issues and the current research status.
- The sample size of this paper is small, which seriously affects the representativeness of the research results for the overall First Nations children in Canada. The authors need to add relevant explanations in the Discussion section.
- Although some factors have been analyzed in the study, there may still be other unconsidered confounding factors that affect sleep patterns, sleep disorders, and excessive daytime sleepiness (EDS). For example, factors like family income, parental education levels, and community cultural activities may be related to children's sleep conditions. The authors should discuss the possible impacts of these potential confounding factors.
- This study focuses on First Nations children. However, in the Discussion section, the analysis of how cultural factors affect sleep habits, sleep disorders, and EDS is not in - depth enough. It is recommended that the authors combine the cultural traditions and lifestyles of the First Nations to explore in more detail the mechanism of the role of cultural factors in children's sleep problems.
- The study was only conducted in one Cree First Nation community in Saskatchewan, Canada. When discussing the significance of the research results, the authors insufficiently discuss their generalizability to other First Nation communities or a broader group of children. The authors should carefully explain the applicable scope of the research results and point out the differences and challenges that may be faced during the generalization process to avoid over - interpreting the research results.
- The authors should put forward more specific intervention suggestions in the Conclusion section.
Author Response
Comments and Suggestions for Authors
This paper focuses on the sleep research of First Nations children in Canada. Although it has certain research value, there are deficiencies in aspects such as relevant experimental considerations, mechanism analysis, and relevant recommendations. My main concerns are as follows:
- The content in the Introduction section is too simplistic. Please conduct an in-depth analysis of the background issues and the current research status.
Response: Thanks for the comment and more details were added to the introduction.
2. The sample size of this paper is small, which seriously affects the representativeness of the research results for the overall First Nations children in Canada. The authors need to add relevant explanations in the Discussion section.
Response: The limitations due to small sample size regarding the generalizability of study findings and reliability of interpretation of results were included into the discussion section. Please see page 13, lines 343-346.
3. Although some factors have been analyzed in the study, there may still be other unconsidered confounding factors that affect sleep patterns, sleep disorders, and excessive daytime sleepiness (EDS). For example, factors like family income, parental education levels, and community cultural activities may be related to children's sleep conditions. The authors should discuss the possible impacts of these potential confounding factors.
Response: Thanks for the comment. In our survey, we have parental highest education level and we included that to the Table 4. However, we did not collect factors like family income and community cultural activities. We realized that they are important factors and included as a limitation and discussed the importance of them. Please see page 13, lines 350-352.
4. This study focuses on First Nations children. However, in the Discussion section, the analysis of how cultural factors affect sleep habits, sleep disorders, and EDS is not in - depth enough. It is recommended that the authors combine the cultural traditions and lifestyles of the First Nations to explore in more detail the mechanism of the role of cultural factors in children's sleep problems.
Response: In our survey, we have only included few questions to cover one cultural practice of “co-sleeping”: When going to sleep or going to bed, how many people are sharing the same room with your child?; Does your child have their own bed? Does your child sleep in the same bed every night?. We have included those variables our analysis and discussed. However, we understand the importance of cultural factors on sleep and discussed in the limitation section. Please see page 13, lines 350-352.
5. The study was only conducted in one Cree First Nation community in Saskatchewan, Canada. When discussing the significance of the research results, the authors insufficiently discuss their generalizability to other First Nation communities or a broader group of children. The authors should carefully explain the applicable scope of the research results and point out the differences and challenges that may be faced during the generalization process to avoid over - interpreting the research results.
Response: As we mentioned earlier in comment #2 due to the small sample size, generalizability was hindered. Therefore, these research results more relevant to the study population rather than the other First Nations communities. Limitation of this was discussed in the discussion. Please see page 13, lines 343-346.
6. The authors should put forward more specific intervention suggestions in the Conclusion section.
Response: The following statements were added to the conclusion section as suggestions for interventions.
“Parents play a crucial role in implementing screen time interventions and establishing bedtime routines for their children. These interventions encourage children to reduce their time spent in front of the TV and other stationary screens, such as computers, video games, and mobile devices like smartphones and tablets. Instead, children should engage in physical activities and have limited screen time. Additionally, parents can benefit from education on how to create healthy bedtime routines. It is also important to minimize noise in the house during bedtime to promote better sleep.” Please see page 16, lines 493-499.
Reviewer 2 Report
Comments and Suggestions for Authors
The study is novel and important. However I found a few flaws:
- The Authors have to imporve the study reporting. I recommend to use the STROBE checklist for your type of observational study.
2. The Introduction is very limited. I recommend to extend the Introduction and report:
- the latest studies related to the sleep of First Nations Children form other regions and countries
- the latest studies which present how sleep disturbances affect children health eg. doi:10.17219/dmp/167411 and doi:10.17219/dmp/150615
- justification why your study is novel and important - please add a paragraph
- please provide accurate information on the prevalence of sleep disorders in children in Canada and several other highly developed countries based on the latest research results
3. Please define clear inclusion and exclusion criteria for study participants.
4. Please add a sample power and size calculation.
5. The Authors have to add information whether the survey was in paper or digital form and how it was distributed. Please provide detailed information regarding the survey distribution process.
Author Response
Comments and Suggestions for Authors
The study is novel and important. However I found a few flaws:
- The Authors have to imporve the study reporting. I recommend to use the STROBE checklist for your type of observational study.
Response: Thanks for the comment and we are already following the “STROBE Statement—Checklist of items that should be included in reports of cross-sectional studies “including sections of Title, Abstract, Introduction (Background/rationale, objectives), Methods(study design, participants, variables, measurements, sample size calculations, statistical methods), Results (descriptive data, outcome data, main results), Discussion (key results, limitations, interpretation and generalizability) and funding details. However, more details were added as requested.
- The Introduction is very limited. I recommend to extend the Introduction and report:
- the latest studies related to the sleep of First Nations Children form other regions and countries
- the latest studies which present how sleep disturbances affect children health eg. doi:10.17219/dmp/167411 and doi:10.17219/dmp/150615
- justification why your study is novel and important - please add a paragraph
- please provide accurate information on the prevalence of sleep disorders in children in Canada and several other highly developed countries based on the latest research results
Response: Thanks for the comment and more details were added to the introduction with related to above points. However, there were not many studies related First Nations Children, but we discussed about Indigenous children from other regions and countries. Please see introduction on page 2.
- Please define clear inclusion and exclusion criteria for study participants.
Response: Inclusion criteria and exclusion criteria were included. “This analysis focuses on data collected from a baseline cross-sectional study conducted in 2024, specifically among children in one Cree First Nation community. All children aged 6-17 years, members of participating reserves and living on reserve and living in the nearby town who were members of the reserve and attending on-reserve schools were eligible to participate and children not attending on-reserve schools will not be eligible to participate in this phase of the research.” Please see page 13, lines 361-366.
- Please add a sample power and size calculation.
Response: A statement of retrospective power calculations was added to the limitations section. Please see page 13, lines 346-350.
“Retrospective power analyses [REF} indicated that this study was moderately powered to detect a difference between the population proportion and the sample proportion of 0.103 (where p0 = 0.300 and p1 = 0.197) in the prevalence of EDS, using a type I error rate (α) of 0.05. A larger sample size may have enhanced the robustness of the results.”
[REF] Rosner, Bernard (Bernard A.). (2011). Fundamentals of biostatistics. 7th ed. Boston :Brooks/Cole, Cengage Learning, 859 p.
- The Authors have to add information whether the survey was in paper or digital form and how it was distributed. Please provide detailed information regarding the survey distribution process.
Response: Survey was in paper form and details of how it was distributed were included in Methods section. Please see page 14, lines 389-397.
Round 2
Reviewer 1 Report
Comments and Suggestions for Authors
Accept
Reviewer 2 Report
Comments and Suggestions for Authors
The manuscript has been correctly revised. I don't have further comments.